# Na_2_S-Mediated One-Pot Selective Deoxygenation of α-Hydroxyl Carbonyl Compounds including Natural Products

**DOI:** 10.3390/molecules27154675

**Published:** 2022-07-22

**Authors:** Xiaobo Xu, Leyu Yan, Zhi-Kai Zhang, Bingqing Lu, Zhuangwen Guo, Mengyue Chen, Zhong-Yan Cao

**Affiliations:** 1College of Chemistry and Pharmaceutical Engineering, Huanghuai University, Zhumadian 463000, China; 20171730@huanghuai.edu.cn (L.Y.); lu2906416834@163.com (B.L.); wen2836118387@163.com (Z.G.); c2725692981@163.com (M.C.); 2College of Chemistry and Chemical Engineering, Henan University, Kaifeng 475004, China; 13781282012@163.com

**Keywords:** deoxygenation, chlorination, dechlorination, one-pot

## Abstract

A practical method for the deoxygenation of α-hydroxyl carbonyl compounds under mild reaction conditions is reported here. The use of cheap and easy-to-handle Na_2_S·9H_2_O as the reductant in the presence of PPh_3_ and *N*-chlorosuccinimide (NCS) enables the selective dehydroxylation of α-hydroxyl carbonyl compounds, including ketones, esters, amides, imides and nitrile groups. The synthetic utility is demonstrated by the late-stage deoxygenation of bioactive molecule and complex natural products.

## 1. Introduction

Deoxybenzoin (DOB) motifs are commonly found in many natural products, pharmaceutically-active molecules and fire-resistant polymers [1,2,3,4]. In addition, some DOB derivatives have been sporadically reported to possess activities such as β estrogenic agonist, antiallergic, anti-inflammatory and antimicrobial activities [5,6,7]. DOBs are industrially prepared from arylacetic acid and arenes by AlCl_3_-catalyzed C–C bond coupling. The process requires the functionalization of phenylacetic acid to phenylacetyl chloride by stoichiometric PCl_3_ or SOCl_2_ prior to the C–C bond coupling [8,9,10,11,12]. Other elegant strategies, including hydration [13], olefin cleavage [14], benzylic oxidation [15] and C–O bond breaking protocols [16,17,18], have also been developed to access DOBs in recent years (Figure 1). However, these methods generally required the prefunctionalization of starting molecules or, alternatively, the use of expensive substrates [13,14,15,16]. Thus, it is highly desirable to develop practical processes for DOB production using cheap and easy-to-handle feedstocks.

On the other hand, benzoins are classically prepared by the cyanide-mediated benzoin condensation of aromatic aldehydes, and, more generally, acyloins have long been efficiently synthesized from esters by the acyloin condensation by using dissolving metals [19,20,21]. Most notably, a wide range of benzoins are commercially available and inexpensive. Therefore, the selective dehydroxylation of benzoins is undoubtedly one of the most powerful and attractive methodologies to access these valuable DOBs products. However, there are currently few methods for directly transforming acyloins to ketones via a dehydroxylation strategy. Moreover, each of the reported methods has limitations, such as the need for metal catalysis, moisture-sensitive reagents, high temperatures, bases, additives or expensive reactants, along with low chemoselectivity or unsatisfactory yields [22,23,24,25,26,27,28,29,30]. In fact, the selective dehydroxylation of such α-hydroxyl carbonyl compounds is nontrivial, as the hydroxyl group is a poor leaving group and the adjacent carbonyl moiety is also susceptible to these reduction conditions. In this context, it is of high interest for developing efficient, mild and economical methodologies for this useful transformation. In view of this, we wish to report a practical and selective one-pot method for the dehydroxylation of benzoins to corresponding DOBs in excellent yields through the in situ chlorination of alcohols and reductive dechlorination using cheap and easy-to-handle PPh_3_/NCS and Na_2_S·9H_2_O as a chlorinated reagent and reductant, respectively.

## 2. Results

Our investigation began with the evaluation of reaction parameters using benzoin (**1a**) as the model substrate (Table 1). Given the cheap and easy-to-handle nature of Na_2_S·9H_2_O, it was chosen as the reductant for our model reaction. After systematically screening the reaction parameters, we found that **1a** could be quantitatively converted to ketone **2a** in the presence of NCS/PPh_3_ at room temperature in one hour when Na_2_S·9H_2_O and DMF were used as the reductant and solvent, respectively (Entry 2). No other side products were formed under the optimized conditions. Three points should be highlighted. (1) The screening of solvents showed that the use of *N*, *N*-dimethylformamide (DMF) is superior, as no improvement of yield was observed when the solvent was switched from DMF to CH_2_Cl_2_, toluene, THF or CH_3_CN (Table 1, Entries 1–5). This might because of the better solubility of Na_2_S·9H_2_O in DMF than in other solvents. (2) When other sulfur-containing reducing agents, such as Na_2_S·5H_2_O, K_2_S, NaSH, NaSH·H_2_O or S_8_, were employed, the desired product **2a** was isolated in relatively lower yields (Entries 6–10). (3) To eliminate the influence of the alkalinity of Na_2_S·9H_2_O on dehalogenation [30], both organic and inorganic bases, including imidazole, pyridine and NaOH, were all investigated, and they all give rise to chloride intermediate instead of DOB **2a** (Entries 11–13).

The preliminary results show that the Na_2_S·9H_2_O as reductant is a good supplement to many of the conventional reductants, such as Zn [22], Sn [23], P [24], P(OEt)_3_ [25] and TMSI [26], for the dehydroxylation of benzoin (Table 2) in terms of the economy and safety of the reagent, as well as the gentleness and efficiency of the reaction.

## 3. Discussion

With an optimized set of reaction conditions established, the scope of dehydroxylation was investigated (Figure 2). The substituents of fluoro, chloro and methoxy at the para position of the benzoyl ring could be well tolerated and they reacted smoothly under the standard conditions, providing the corresponding DOB products with 88–95% yields (**1b**–**1d**). Similarly, the introduction of either electron-withdrawing or electron-donating substituents on the phenyl ring did not alter the reaction efficiency as demonstrated by the chloro and methyl substituents (**1f**–**1g**). To our delight, one representative heteroaromatic furan-derived 1e was well tolerated enough to afford the corresponding product **2e** a 84% yield. Moreover, the dehydroxylation of **1h** and **1i** bearing two substituents on the benzoyl and phenyl rings also worked efficiently.

Moreover, our dehydroxylation strategy could be scaled up the Gram-scale smoothly, providing a new and practical way for the synthesis of high value-added ketone **2h** from the cheap substrate **1h** at a 86% yield under mild conditions (Figure 3). The price of ketone **2h** is 38 times higher than that of the start material [31].

To avoid the problem of the use of stoichiometric Ph_3_P possibly causing the tedious separation of the phosphine-derived byproduct from the desired products, a modified one-pot procedure which includes the triphenylphosphine oxide-catalyzed chlorination reaction of the alcohol **1a** to afford chloride [32] and then dechlorination using Na_2_S·9H_2_O as reductant in MeOH has been established. As shown in Figure 4, this modified and atom-efficient procedure provides a convenient purification, delivering the product at a good yield.

To further explore the scope of our system, other types of α-hydroxyl carbonyl compounds have also been evaluated (Figure 5). Firstly, the primary alcohol (**1k**) in positions α of a ketone group under our conditions reacted well, yielding the corresponding acetophenone **2k** in a moderate yield. Unexpectedly, the secondary alcohols (**1l** and **1m**) also facilitated this transformation, more efficiently than that of primary alcohol (**1k**) under same conditions. In addition, the tertiary alcohol (**1r**) could also be converted to the corresponding dehydroxylation product at a 73% yield, which indicates that the steric effect of substituents in the α-positions of a ketone group had a marginal influence on the yield. Aside from simple phenylacetyl group (**1l**), a broad range of α-hydroxyl carbonyl compounds bearing the aliphatic (**1o**), cyclic (**1p**) and dicarbonyl groups (**1q**) also reacted smoothly.

Pleasingly, the desired dehydroxylation strategy could be extended to other carbonyl-based electron-withdrawing groups, including the ester (**1s**), amide (**1t**) and imide groups (**1u**), as shown by the conversion of these commercially available start materials to give the desired products in good yields (73–87%). In the case of substrate bearing a nitrile group (**1v**) [33,34], the reaction system afforded the dehydroxylation product efficiently, albeit with the requirement of a relatively longer reaction time. Interestingly, as shown in the conversion of a *trans*-3,4-dihydroxypyrrolidine-2,5-dione derivative **1u** into the corresponding **2u**, double dehydroxylation was possible by using the 2.0 equiv. of NCS/PPh_3_ and Na_2_S·9H_2_O.

To further demonstrate the synthetic value of our methodology, the dehydroxylation protocol has been applied for the synthesis of bioactive molecules and the late-stage modification of natural products (Figure 6). For example, flavanone is a natural plant flavonoid found to inhibit tumor cells in vitro [35,36]. The 3-hydroxyflavanone **1w** could be easily transformed into flavanone under our standard reaction conditions. Additionally, cortexolone **1x** could be deoxygenated in a selective manner without affecting the tertiary hydroxyl group. The latter case represents an advantage over the competing SmI_2_-mediated dehydroxylation reaction [28], as the enone moiety is compatible in our case. These examples further demonstrated that our strategy represents an efficient and versatile method for the dehydroxylation of α-hydroxyl carbonyl compounds under mild conditions.

In order to confirm the role of Na_2_S·9H_2_O, substituted acetophenones bearing various leaving groups at the α-position have been evaluated. As shown in Figure 7, benzoin derivatives bearing chloro (**3a**), bromo (**3b**) or methanesulfonate (**3c**) groups at the α-position all reacted smoothly under the standard conditions, providing DOB **2a** at 91–98% yields. These results demonstrate that α-chloro acetophenone might be the plausible intermediate. The use of air atmosphere or adding a radical scavenger, such as 2,2,6,6-tetramethyl-1-piperidinyloxy (TEMPO), to the reaction had almost no effect on the yield. Considering that the reaction could work smoothly in air or with TEMPO, it seems unlikely that the radical process might be involved in our transformation. Furthermore, when the load of Na_2_S·9H_2_O was decreased to 0.5 equiv., the reaction could also proceed smoothly to give **2a** at an 85% yield. As a comparison, the use of BnCl as the substrate only led to the isolation of BnSBn under our standard reaction conditions, indicating the essential role of the α-carbonyl group for activating the substrate for the reaction. Apparently, further studies are necessary to shed light on the reaction mechanism.

Moreover, an α-chloroacetophenone-bearing phenylsulfonyl (**3d**) group proved to be a competent substrate, affording the desired dechlorination product **2j** at an 82% yield under the standard reaction conditions. These results revealed that the leaving group at the α-position of the carbonyl compounds was not limited to Cl, others such as Br- and OM-substituted analogues also worked well in our hand.

## 4. Materials and Methods

Unless otherwise noted, the reactions were carried out in oven-dried glassware or a sealed tube under ambient atmosphere. *N*, *N*-Dimethylformamide (DMF) was distilled from calcium hydride. Tetrahydrofuran (THF) was dried and distilled from sodium. Reactions were monitored by analytical thin-layer chromatography (TLC) on Merck silica gel 60 F_254_ plates (0.25 mm), visualized by ultraviolet light (254 nm) or by staining with ceric ammonium molybdate. ^1^H NMR spectra were obtained on a Bruker AVANCE 400 MHz spectrometer at ambient temperature. Data were reported as follows: chemical shift on the δ scale using residual proton solvent as internal standard [δ TMS: 0.00 ppm], multiplicity (s = singlet, d = doublet, t = triplet, q = quartet, m = multiplet, dd = doublet of doublets), integration and coupling constant (*J*) in hertz (Hz). ^13^C NMR spectra were obtained with proton decoupling on a Bruker AVANCE (100 MHz) spectrometer and were reported in ppm with residual solvent for internal standard [δ 77.0 (CHCl_3_)].

## 5. Conclusions

In summary, an efficient and mild method for the selective dehydroxylation of α-hydroxyl carbonyl compounds was developed using a one-pot strategy, which includes the successive chlorination and reductive dechlorination with NCS/PPh_3_ and Na_2_S·9H_2_O, respectively. The easy-to-handle protocol provides facile, rapid and chemoselective access to DOBs at room temperature without the need for hazardous reagents or expensive metals. The synthetic utility of the methodology has been demonstrated by the facile synthesis of the bioactive molecule, the late-stage dehydroxylation of the complex natural product and Gram-scale transformation into a high value-added chemical.

## Data Availability

Data are contained within the article or Appendix A.

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
