# Peer review of "Na2S-Mediated One-Pot Selective Deoxygenation of α-Hydroxyl Carbonyl Compounds including Natural Products"

_molecules, 2022, doi:10.3390/molecules27154675_

Round 1

Reviewer 1 Report

The paper describes efficient one-pot deoxygenation of alfa-hydroxyl carbonyl compounds mediated by NCS/PPh3 and Na2S·9H2O. The conditions are applicable to wide range of alfa-hydroxyl carbonyl compounds including ketones, esters, amides, as well as cyanohydrin. Although the whole reaction mechanism remains unsolved, alcohol-chlorination is thought to be involved in the first step and Na2S·9H2O facilitates de-chlorination in the second step without involving the radical pathway. The good point is that not only chlorides but also bromide and mesylate can be converted to the reduced product by Na2S·9H2O. The reaction conditions are very simple and useful for the preparation of a variety of alfa-deoxygenated carbonyl compounds from the hydroxylated substrates. The reviewer is positive to publish this paper to Molecules because of the usefulness in broad synthetic communities.

The negative point is that “No experimental procedures and spectral data are included in text” although authors show general techniques in the section 4. Material and Methods. The reviewer suspects that the authors have uploaded an unfinished manuscript.

Other suggestions are listed below.

24: antimicrobial activities.

Scheme 1: Vertical lines are not necessary. SmI2, TMSI, etc.

37: Why did authors add “(alfa-hydroxyketones and aldehydes)” here?

46: Delete “et al”

48: “...carbonyl moiety also tends to be reduced…” or “...carbonyl moiety is also susceptible to these reduction conditions.”

58: Na2S·9H2O (add 9)

63: Screening of solvents…

66: DMF than in other solvents

67: NaSH·H2O

68: Delete “as the reductant”.

70: imidazole

Table 1: Delete all “%” in entries. (c) …was “detected (?)” by GC-MS

85: demonstrated by the

Scheme 2: 136 RMB/1 g, 5125 RMB/1 g

105-106: …tedious separation “of Ph3PO (?)” from… or “phosphine-derived byproduct (?)”

131: shown in the conversion

132: dihydroxylation was possible by using

145: Add reference after “in vitro”.

148: The latter case respresents…

157: Table 4

161: The following sentence “These results…plausible intermediate” should be inserted after 91-98% yield in line 159.

189: [δ 7.26 (CHCl3); TMS: 0.00 ppm] …. Double standard is not acceptable.

200: the bioactive

201: of the complex

202: into a high value-added chemical.

Acknowledments: Should eliminate funding information? (overlap)

276: 136 RMB/1 g, 5125 RMB/1 g

Reviewer 2 Report

1.      Deoxygenation of α-hydroxy Carbonyl Compounds also can be done by powdered tin in large scale. Please see: Org. Synth. 196040, 16. The authors should be mentioned this method and others such as aqueous HClO4 in the introduction.

2.      Please insert superiority of Na2S over some reported reductant for a model reaction in a Table based on mol% of catalyst, time of the reaction, temperature, yield,...

3.      There is no data about 1HNMR and 13CNMR of the products. This is a defective point of the work. Please prepare the copies of 1HNMR and 13CNMR of the products in the supporting information file.

4.      Relevant references of the synthesized compounds should be cited.

5.      Please added some descriptive text about spectral data of the products to the manuscript. The authors have declared in the Materials and Methods section the NMR instrument specifications; but no data in the manuscript and not supporting information file.

6.      The authors have declared that they prepared supporting information file “The following supporting information can be downloaded at: www.mdpi.com/xxx/s1, Figure S1: title; Table S1: title; Video S1: title.”. I cannot find it. What is this?

7.      There is no data about mass spectral data of the new products. This is also a defective point of the work.

8.      In Scheme 4, how you can confirm the structure of the product 2x? Maybe another hydroxyl group undergoes the reaction? Mono dehydroxylation or double dehydroxylation? Which one?

9.      In Table 2, the authors should be inserted structure of products.

10.  In Table 3, the authors should be inserted structure of products. What is the product in the case of 1u? Mono dehydroxylation or double dehydroxylation?

11.  The mechanism of reaction should give. Insert the some experiments to confirm the proposed mechanism. What is the plausible role for Na2S in the reaction?

12.  What is the result of the reaction of α-diketones with the reported reductive conditions?

Round 2

Reviewer 2 Report

Dear Editor
The paper was revised according to the reviewer’ comments.
In its current state it is ready for publication in your journal.
Best regards